# DualCL: Principled Supervised Contrastive Learning as Mutual Information Maximization for Text Classification

## ABSTRACT

Text classification is a fundamental task in web content mining. Developing text classification applications with pre-trained language models (PLMs) and the contrastive learning objective has sparked significant interest in research communities. Although the existing supervised contrastive learning (SCL) approach has achieved leading performance in text classification, it lacks fundamental principles to ensure training effectiveness and deployment friendliness, thereby presenting certain limitations. In this paper, we propose three principles to design an effective SCL approach, i.e., parameter-free, augmentation-easy and label-aware. Building upon these principles, we have developed DualCL, a dual contrastive learning framework that effectively captures the mutual relationship between text representations and classifier parameters. The implementation of DualCL is theoretically motivated by a derived lower bound of mutual information maximization. DualCL generates classifier parameters by the PLM and simultaneously uses them for classification and as augmented views of the input text for supervised contrastive learning. Extensive experiments conducted on diverse text classification datasets conclusively demonstrate that DualCL excels in learning superior text representations and consistently outperforms baseline models, yielding remarkable results.

## CCS CONCEPTS

• **Computing methodologies → Natural language processing**.

## KEYWORDS

Text Classification, Mutual Information, Contrastive Learning

**ACM Reference Format:**
Anonymous Author(s). 2018. DualCL: Principled Supervised Contrastive Learning as Mutual Information Maximization for Text Classification. In *Proceedings of Make sure to enter the correct conference title from your rights confirmation emai (Conference acronym 'XX)*. ACM, New York, NY, USA, 10 pages. https://doi.org/XXXXXXX.XXXXXXX

## 1 INTRODUCTION

Text classification plays a pivotal role in web content mining, which serves as a crucial component in a diverse array of web applications, such as sentiment analysis [27], review classification [23] and question classification [20]. Pretrained language models (PLMs) have demonstrated stunning successes in text classification tasks

**Figure 1: Existing SCL framework (a) and our DualCL framework (b). The former requires learning additional classifiers. And the latter mutually regard text representations and class parameters as anchors for contrastive learning.**

by learning informative representations [6, 16]. The representation power of PLMs is recently invigorated by *contrastive learning*, an effective representation learning approach that aims to force the representations of matched examples to be similar and the representations of unmatched examples to be distinct.

A primary attempt to adopt contrastive learning in text classification [10, 36, 39] is inspired by the idea of obtaining generic representations for downstream tasks [3, 14] using unsupervised contrastive learning. These methods promote the proximity of representations from various *views* of the same example while ensuring that representations of different examples remain distant. Although shown promising results, unsupervised contrastive learning, not surprisingly, still has a yawning gap with supervised learning approaches. Therefore, supervised contrastive learning (SCL) [17] has been proposed to extend the training strategy of unsupervised contrastive learning to a situation with supervision signals, which pulls the representations of examples with the same label closer and pushes the representations of examples with different labels away. It is applied in a series of recent works and demonstrated to be effective in different text classification scenarios [2, 4, 12, 28, 30]. However, the existing SCL approach, which is primarily derived from the principles of unsupervised contrastive learning, may benefit from further refinement to better accommodate the unique properties associated with the supervised setting.

In this paper, we attempt to shed light on the potential shortcomings of the existing SCL approach and try to develop a principled SCL approach for text classification. Our investigation reveals that the following three factors may impact the training effect and deployment of the existing SCL models. *First,* as shown in Figure 1 (a), the outcome of most existing approaches does not give a classifier directly. Therefore, existing models train the supervised contrastive learning loss and a softmax classifier successively [17] or jointly [12] to perform classification. These models require learning classifiers with additional parameters, increasing the deployment workload.

*Second,* some existing SCL methods require external text data augmentation techniques [1, 26, 32, 33] to obtain different views of the examples. Building independent modules for data augmentation complicates the training process. *Third,* most existing SCL models only treat the labels as evidence to construct matched and unmatched pairs [12, 17], ignoring underlying semantics of the classes that may encourage learning good text representations.

Based on the above discussion, we propose the following three principles to overcome the shortcomings of the existing SCL approach. **Parameter-Free:** The SCL models should perform classification without learning classifiers with additional parameters. **Augmentation-Easy:** The SCL models should easily obtain different views of representations without external data augmentation techniques. **Class-Aware:** The SCL models should sufficiently tap the potential of class semantics for learning better representations. Developing models following the above principles helps to outcome training-effective and deployment-friendly SCL models.

As shown in Figure 1 (b), we intend to achieve parameter-free SCL by *generating* the parameters of the classifier using the PLM encoder shared with the representation learning module instead of training an extra classifier. In this way, the classification is performed without introducing additional parameters. To realize augmentation-easy and class-aware SCL, the generated classifier parameters corresponding to different classes are treated as different augmented views of the text. Then, the multi-view representations are leveraged in supervised contrastive learning. This strategy allows us to easily obtain the augmentations of texts from *internal* components, releasing the SCL model from external data augmentation techniques. Additionally, leveraging classifier parameters as different views of data augmentations effectively incorporates class semantics, leading to enhanced representation learning.

To follow up the above design, we interpret the duality between text representations and classifier parameters and theoretically derive a lower bound of mutual information maximization. Guided by the theoretical findings, we develop a supervised contrastive learning framework that directly generates classifier parameters for each input from the PLM and uses them as augmented views of the input text during representation learning. Concretely, on the one hand, text representations and the generated classifier parameters are mutually regarded as anchors and different views of positive/negative examples to conduct supervised contrastive learning. On the other hand, the generated classifier parameters are concatenated as the transformation matrix of the linear classifier to perform softmax classification. Our method generates a classifier for each input text, enabling us to aggregate these classifiers into a more robust and effective classifier by the ensemble approach. We christen our framework as Dual Contrastive Learning (DualCL).

To summarize, this work makes the following contributions: (1) We reveal the shortcomings of the existing SCL approach and put forward three key principles - parameter-free, augmentation-easy, and class-aware - to construct a more effective SCL model. (2) We theoretically interpret the duality property of text classification with mutual information maximization and accordingly present a dual contrastive learning framework DualCL for text classification. (3) We conduct experiments on 5 datasets, demonstrating that our DualCL framework is capable of learning improved text representations and achieving superior performance compared to baselines.

## 2 RELATED WORK

### 2.0.1 Text Classification.
Text classification plays a fundamental role in natural language processing, serving as a vital tool for web content mining. It enables the categorization of texts into distinct groups based on their underlying semantics. Text classification has a wide range of applications, including sentiment analysis [27], question answering [20], etc. Extracting meaningful representations from text can be challenging due to its inherent unstructured nature. The advent of deep learning has brought about significant advancements in the field, leading to extensive exploration of neural network methods like Recurrent Neural Networks (RNN) [5, 15] and Convolutional Neural Networks (CNN) [18, 35] for effective encoding of text sequences. However, these methods face certain limitations, including computational bottlenecks and the challenge of capturing long-term dependencies within the text.

Recently, large-scale pre-trained language models (PLMs) based on transformers [29] have emerged as the state-of-the-art approach for text modelling. These PLMs, such as GPT [25], XLNet [34], BERT [6], RoBERTa [21], and ALBERT [19], have made significant strides in advancing text classification tasks. Auto-regressive PLMs, like GPT and XLNet, leverage sequential generation, while auto-encoding PLMs, including BERT, RoBERTa, and ALBERT, focus on sentence-level representation. These models have demonstrated remarkable improvements in text classification performance.

### 2.0.2 Contrastive Learning.
Despite the widespread use of cross-entropy in supervised learning, several studies have shed light on its limitations. For instance, it has been found to be susceptible to issues such as vulnerable to noisy labels [37], poor margins [8], and weak adversarial robustness [24]. These drawbacks have prompted researchers to explore alternative loss functions and techniques to address these challenges. Drawing inspiration from the InfoNCE loss [22], contrastive learning [13] has gained significant popularity in unsupervised learning as a means to obtain high-quality generic representations for downstream tasks. One notable application is SimCLR [3], which generates multiple views of input examples by applying data augmentations and subsequently compares positive samples against negative samples within the dataset. Similarly, SimCSE [11] employs dropout on each sentence twice to create positive pairs. These approaches leverage contrastive learning to enhance representation learning in an unsupervised setting.

Supervised contrastive learning [17] builds upon the principles of unsupervised contrastive learning, extending its application to a supervised setting. This methodology has been successfully employed in various text classification scenarios [2, 4, 12, 28, 30]. Building a class-aware SCL approach is not entirely unknown to the research community. A recent work LaCon [38] considers label information in SCL and explores the interplay between text and label representations. The primary distinction between DualCL and LaCon is twofold. First, LaCon introduces a learnable matrix with extra parameters to look up label embeddings, while DualCL directly generates classifier parameters from PLM for each input, following the parameter-free principle. Second, LaCon is theoretically motivated by the complementarity of contrastive losses from the perspective of the singular values, while DualCL is theoretically motivated by the duality between text representations and classifier parameters from the perspective of mutual information maximization.

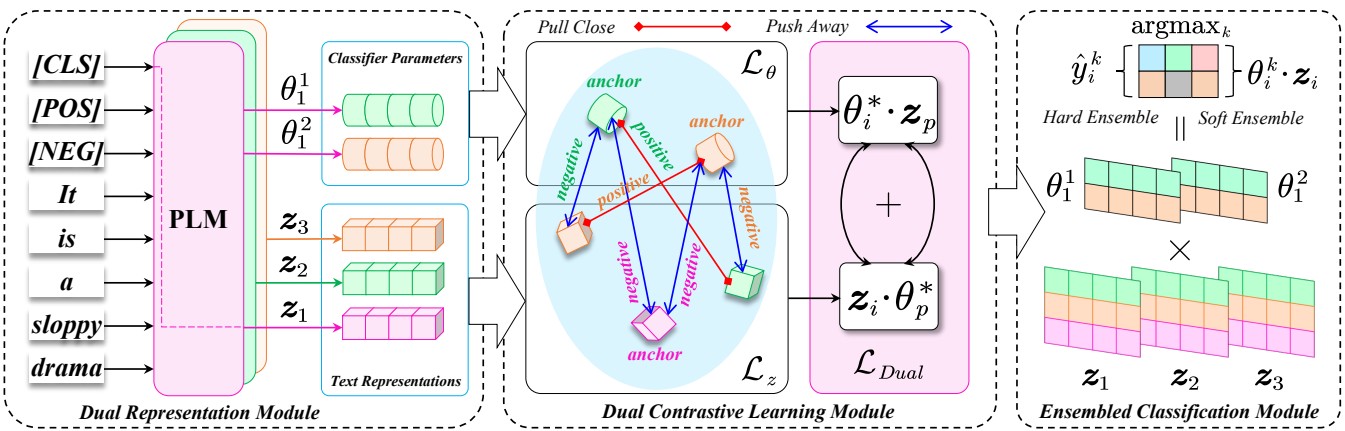

Figure 2: The framework of the proposed dual contrastive learning (DualCL).

## 3 PRELIMINARIES

### 3.1 Task Definition

The text classification task is concerned with predicting the class or category of a given text, encompassing various applications such as sentiment analysis, review classification, and question classification. Formally, let's consider a text classification task with $K$ classes. The input-text space is denoted as $\mathcal{X}$, and the label space is $\mathcal{Y}$. We assume that the given dataset $\mathcal{D} = \{x_i, y_i\}_{i=1}^N$ comprises $N$ training examples, where $x_i \in \mathcal{X}$ is the text-token sequence with $L$ words, and $y_i \in \{1, 2, \cdots, K\}$ represents the assigned class label for the input text. Throughout this study, our focus lies on text classification using representation learning methods. Specifically, the input text $x_i$ is initially processed by an encoder $f$, which produces a representation $z_i \in \mathbb{R}^d$, denoted as $f(x_i) = z_i$. Subsequently, classification is conducted based on the obtained representation $z_i$.

### 3.2 Supervised Contrastive Learning

Previous studies utilize unsupervised or self-supervised contrastive learning for text classification [10, 36]. Since these methods lack supervision signals, they suffer a performance gap compared to supervised approaches. In order to bridge this gap, supervised contrastive learning (SCL) [17] has recently emerged, which extends the principles of contrastive learning to a supervised setting. SCL aims to enhance the discriminative power of learned representations by enforcing the representations of instances belonging to the same class to be closer, while simultaneously pushing apart the representations of instances from different classes.

Specifically, let $\mathcal{I} := \{1, 2, \cdots, N\}$ be the set of indexes of the input texts. $\mathcal{A}_i := \mathcal{I} \setminus \{i\}$ is the set of indexes of all *positive* and *negative* examples when the $i^{\text{th}}$ representation $z_i$ acts as an *anchor*. Given dataset $\mathcal{D}$, the supervised contrastive learning loss is

$$\mathcal{L}_{\text{sup}} = \frac{1}{N} \sum_{i \in \mathcal{I}} \frac{1}{|\mathcal{P}_i|} \sum_{p \in \mathcal{P}_i} -\log \frac{\exp(z_i \cdot z_p / \tau)}{\sum_{a \in \mathcal{A}_i} \exp(z_i \cdot z_a / \tau)}, \quad (1)$$

where $\mathcal{P}_i := \{p \in \mathcal{A}_i : y_p = y_i\}$ is the set of indexes of positive examples, and $|\mathcal{P}_i|$ is the cardinality of $\mathcal{P}_i$. The $\cdot$ symbol denotes the inner product and the parameter $\tau$ is the temperature factor.

Despite the successful application of supervised contrastive learning (SCL) in various text classification scenarios [2, 12], this approach may suffer certain shortcomings. Firstly, it requires learning an extra classifier, which introduces additional parameters to be trained. Secondly, SCL relies on external text data augmentation techniques, which can be computationally expensive and may not always be readily available or applicable to all datasets. Lastly, SCL overlooks the incorporation of label semantics into the representation learning process, which could potentially enhance the quality of learned representations. The existing SCL approaches can be refined by effectively overcoming the above limitations.

## 4 METHODOLOGY

To mitigate the shortcomings of the existing SCL approach, we propose three principles to design more effective SCL models, i.e., parameter-free, augmentation-easy and class-aware. Laying the foundation on the principles, we design a dual contrastive learning framework, DualCL, motivated by a derived lower bound of *mutual information maximization*. DualCL comprises three components: *Dual Representation Module*, *Dual Contrastive Learning Module* and *Ensembled Classification Module*, as the framework shown in Figure 2. This section will begin by outlining the theoretical motivation behind DualCL and subsequently present its implementation.

### 4.1 Theoretical Motivation

*4.1.1 The Duality between Text Representations and Classifier Parameters in Text Classification.* Considering solving the text classification problem with a linear classifier. Let $z_i \in \mathbb{R}^d$ be the representations of text $x_i$ and $\theta \in \mathbb{R}^{d \times K}$ be the parameters of the classifier. Let $\theta^*$ are the classifier parameters associated with the ground-truth label. When training this model, the softmax transformation of inner product $\theta^* \cdot z_i$ is forced to be maximized through the optimizer. This forms a dual relationship between the text representations and the classifier parameters. When we adopt the inner product in representation learning, it shows similar duality between different representations. This motivates us to consider: *Can we enhance text representation learning with such intrinsic* **duality** *property?*

*4.1.2 Rethink the Duality form A Mutual Information Maximization Perspective.* To answer the question, we delve into understanding the duality in text classification from the perspective of mutual information (MI) maximization. MI quantifies the dependence between the input features and class labels, which helps identify the most informative features strongly associated with specific classes. The symmetrical nature of MI enables a seamless connection to the duality between text representations and classifier parameters. And this opportunity presents a captivating avenue for maximizing MI by leveraging the duality property in text classification.

Concretely, let $\mathcal{X} = \{x_i\}_{i=1}^N$ and $\mathcal{Y} = \{y_i\}_{i=1}^N$ respectively denote the input texts and class labels of $N$ samples in the training dataset and $\text{MI}(\mathcal{X}, \mathcal{Y})$ denotes mutual information between $\mathcal{X}$ and $\mathcal{Y}$. Assume that each example $x_i$ is encoded to a representation $z_i$ and is assigned a specific classifier with parameters $\theta_i$, referred to as *one-example classifier*, which will be further elaborated. And we denote the part of parameters corresponding to ground-truth label $y_i$ as $\theta_i^*$. Then, we can define a symmetric function as follows

$$\phi(x_i, y_j) = (g(\theta_i^*, z_j) + g(\theta_j^*, z_i))/2 \quad (2)$$

With the above-defined function, we derive a lower bound of mutual information $\text{MI}(\mathcal{X}, \mathcal{Y})$ that forms the following theorem

THEOREM 1. *Assume that there is a constant $\epsilon$ such that $p(x_i, y_i) \geq \epsilon$ holds for all $i \in \mathcal{I}$ and $\frac{p(y_j|x_i)}{p(y_j)} \propto \phi(x_i, y_j)$:*

$$
\begin{aligned}
\text{MI}(\mathcal{X}, \mathcal{Y}) &= \frac{1}{N} \sum_{i=1}^N \sum_{j=1}^N p(x_i, y_j) \log\left(\frac{p(y_j|x_i)}{p(y_j)}\right) \\
&= \frac{1}{N} p(x_i, y_i) \log \frac{\phi(x_i, y_i)}{\sum_{t=1}^N \phi(x_i, y_t)} \\
&\geq \log N + \frac{\epsilon}{N} \sum_{i=1}^N \log \frac{\phi(x_i, y_i)}{\sum_{t=1}^N \phi(x_i, y_t)} \\
&\geq \log N + \frac{\epsilon}{N} \sum_{i=1}^N \frac{1}{|\mathcal{P}_i|} \sum_{p \in \mathcal{P}_i} \log \frac{g(\theta_i^*, z_p)}{\sum_{a=1}^N g(\theta_i^*, z_a)} \\
&\quad + \frac{\epsilon}{N} \sum_{i=1}^N \frac{1}{|\mathcal{P}_i|} \sum_{p \in \mathcal{P}_i} \log \frac{g(\theta_p^*, z_i)}{\sum_{a=1}^N g(\theta_a^*, z_i)} \\
&= \log N - \epsilon(\mathcal{L}_{\theta \leftarrow z} + \mathcal{L}_{z \leftarrow \theta})
\end{aligned}
\quad (3)
$$

*where $\mathcal{P}_i$ is a text set associated with $x_i$, which is sampled from $\mathcal{X}$.*

We give the detailed proof of Theorem 1 in *Appendix*. This theorem indicates that the negative of the summed loss $\mathcal{L}_{\theta \leftarrow z} + \mathcal{L}_{z \leftarrow \theta}$ serves as a lower bound of the mutual information. As a result, minimizing the loss $\mathcal{L}_{\theta \leftarrow z} + \mathcal{L}_{z \leftarrow \theta}$ is equivalent to maximizing mutual information between the inputs and labels. Next, we will describe how to implement this loss with our DualCL framework.

## 4.2 Dual Representation Module

*4.2.1 The Necessity of Generating Classifier Parameters.* Theorem 1 reveals that classifier parameters can play essential roles in learning informative representations and we need construct a one-example classifier for each input text to exploit the duality nature of text classification. However, creating learnable classifiers for all input

texts is crude and impractical. Fortunately, in line with the principle of *parameter-free* SCL, we discover an elegant approach that entails generating classifier parameters directly through the PLM. This obviates the requirement for additional parameters to build the classifier. The generated classifier parameters can be regarded as augmented views of the input associated with certain classes, eliminating the need for external data augmentation techniques.

*4.2.2 Generating Text Representations and Classifier Parameters.* To obtain dual representations of the texts and class parameters, we utilize PLMs as the encoder $f$. Each class label is associated with a special token and forms a class-token sequence. For instance, we can use "[POS]" and "[NEG]" to represent the positive and negative classes in sentiment classification, respectively. Meanwhile, we concatenate the class-token and text-token sequence and add a special token "[CLS]" at the beginning. This forms a new sequence $r_i \in \mathbb{R}^{K+L+1}$. Then, we pass the sequence $r_i$ through the PLM encoder $f$. The encoder transforms the input sequence into $K + L + 1$ hidden representations. The hidden representation corresponding to the "[CLS]" token is treated as the text representation $z_i$. The hidden representation for each token in the class-token sequence is accordingly treated as the classifier parameter $\theta_i^k$, where $k$ represents the index of the token in the class-token sequence. This design allows the classifier parameters to be generated directly from the PLM, eliminating the need for introducing additional parameters and providing convenience for *augmentation-easy* implementation.

## 4.3 Dual Contrastive Learning Module

*4.3.1 Dual Contrastive Loss.* Motivated by the form of the MI lower bound in Theorem 1, we intend to build a pair of losses that satisfies duality. With the text representation $z_i$ and the classifier parameters $\theta_i$, we attempt to align the softmax transforms of $\theta_i^T z_i$ with the label of $x_i$. Namely, let $\theta_i^*$ denote the column of $\theta_i$ corresponding to the ground-truth label, then we expect the inner product $\theta_i^* \cdot z_i$ to be maximized. To that end, we design a dual contrastive loss to model the dual relationship between text representations and classifier parameters by mutually regarding them as anchors and different views of positive/negative examples to perform SCL.

Concretely, given an anchor $z_i$ originating from the input example $x_i$, we take $\{\theta_j^*\}_{j \in \mathcal{P}_i}$ as positive examples and $\{\theta_j^*\}_{j \in \mathcal{A}_i \setminus \mathcal{P}_i}$ as negative examples and define the following contrastive loss

$$\mathcal{L}_z = \frac{1}{N} \sum_{i \in \mathcal{I}} \frac{1}{|\mathcal{P}_i|} \sum_{p \in \mathcal{P}_i} -\log \frac{\exp(\theta_p^* \cdot z_i / \tau)}{\sum_{a \in \mathcal{A}_i} \exp(\theta_a^* \cdot z_i / \tau)} \quad (4)$$

Similarly, given an anchor $\theta_i^*$, we can also take set $\{z_j\}_{j \in \mathcal{P}_i}$ as positive examples and $\{z_j\}_{j \in \mathcal{A}_i \setminus \mathcal{P}_i}$ as negative examples. Then, we can define another contrastive loss as follows

$$\mathcal{L}_\theta = \frac{1}{N} \sum_{i \in \mathcal{I}} \frac{1}{|\mathcal{P}_i|} \sum_{p \in \mathcal{P}_i} -\log \frac{\exp(\theta_i^* \cdot z_p / \tau)}{\sum_{a \in \mathcal{A}_i} \exp(\theta_i^* \cdot z_a / \tau)} \quad (5)$$

We combine the two losses $\mathcal{L}_z$ and $\mathcal{L}_\theta$ as the following dual contrastive learning loss

$$\mathcal{L}_{\text{Dual}} = \mathcal{L}_z + \mathcal{L}_\theta \quad (6)$$

The dual contrastive loss leverages class parameters with class semantics infused in contrastive learning, enabling *class-aware* optimization that helps learn better representations.

### 4.3.2 The Equivalence between Minimizing DualCL and Maximizing MI.
By connecting Equation (4) and Equation (5) with the derivation (3) in Theorem 1, we can conclude that the dual contrastive learning loss is exactly equal to the lower bound $\mathcal{L}_{\theta \leftarrow z} + \mathcal{L}_{z \leftarrow \theta}$ of the mutual information $\text{MI}(\mathcal{X}, \mathcal{Y})$ with the function $g$ specified as

$$g(\theta_i^*, z_j) = \exp(\theta_i^* \cdot z_j / \tau) \tag{7}$$

Therefore, minimizing the dual contrastive loss $\mathcal{L}_{\text{Dual}}$ can accordingly maximize MI between inputs and labels, thus helping learn informative representations for text classification.

## 4.4 Ensembled Classification Module

### 4.4.1 Model Training and Testing.
To fully exploit the supervised signal, we also expect the generated parameters $\theta_i$ to be good data augmentations for $z_i$. Thus, we also use a modified version of the cross-entropy loss to maximize $\theta_i^* \cdot z_i$ for each input example $x_i$

$$\mathcal{L}_{\text{CE}} = \frac{1}{N} \sum_{i \in \mathcal{I}} - \log \frac{\exp(\theta_i^* \cdot z_i)}{\sum_{k \in \mathcal{K}} \exp(\theta_i^k \cdot z_i)} \tag{8}$$

We jointly optimize the two objectives to train encoder $f$, which simultaneously improves the representations and classifier parameters by the following loss encompassing both objectives

$$\mathcal{L}_{\text{overall}} = \mathcal{L}_{\text{CE}} + \lambda \mathcal{L}_{\text{Dual}} \tag{9}$$

where $\lambda$ is a hyperparameter to control the loss weight. We name the model trained with this loss **DualCL**.

During the testing phase, we utilize the trained encoder $f$ to generate a text representation $z_i$ and a classifier $\theta_i$ for each input example $x_i$. Here, $\theta_i$ can be seen as a one-example classifier that is specific to the example $x_i$. We regard the label with the maximum value of $\theta_i^k \cdot z_i$ as the predicted label, formulated as follows

$$\hat{y}_i = \arg\max_k (\theta_i^k \cdot z_i) \tag{10}$$

### 4.4.2 Ensembled Classification.
Suppose that the test set $\{x_i\}_{i=1}^M$ consists of $M$ examples. During testing with DualCL, for each input text $x_i$, a one-example classifier $\theta_i$ is generated for classification. Namely, we can generate $M$ classifiers $\{\theta_i\}_{i=1}^M$ by sending all test examples to encoder $f$. These classifiers can be viewed as "weak" classifiers of the classification task, and we can adopt the ensemble method to aggregate the classifiers as a "strong" classifier. In this context, we introduce two ensembled classification methods to combine the outputs of the individual classifiers $\{\theta_i\}_{i=1}^M$.

The first one is Hard Ensembling (HE). This method first predicts $M$ labels for a test example $x_i$ using each classifier in $\{\theta_i\}_{i=1}^M$, then conducts a majority vote on the $M$ labels and treats the most frequent one as the final predicted label. We name the DualCL model using the hard ensembling method in prediction **DualCL+HE**. The second one is Soft Ensembling (SE). For each test example $x_i$, this method first computes the softmax transform scores $\theta_i^T z_i$ for all classifier parameters $\{\theta_i\}_{i=1}^M$, then averages the $M$ score vectors and treats the label corresponding to the maximum score as the final predicted label. We call the DualCL model adopting soft ensembling in prediction **DualCL+SE**. Note that the process of classifier ensembling can be efficiently achieved through matrix multiplication, which greatly reduces the computational complexity involved.

**Table 1: The statistics of the five text classification datasets. #Class, AvgLen, #Train and #Test denote the number of classes, the average length of the texts, the size of the training set and the size of the test set, respectively.**

| Dataset | #Class | AvgLen | #Train | #Test |
|---------|--------|--------|--------|-------|
| SST-2 | 2 | 17 | 7,447 | 1,821 |
| SUBJ | 2 | 21 | 9,000 | 1,000 |
| TREC | 6 | 9 | 5,452 | 500 |
| PC | 2 | 7 | 32,097 | 13,759 |
| CR | 2 | 18 | 3,394 | 376 |

## 5 EXPERIMENTS

## 5.1 Datasets and Experiment Settings

### 5.1.1 Datasets.
We evaluate our models on 5 benchmark text classification datasets: *SST-2*, *SUBJ*, *TREC*, *PC* and *CR*. SST-2 [27] is a sentiment classification dataset of movie reviews. The SUBJ dataset [23] contains labeled sentences categorized as either subjective or objective. It serves as a benchmark for tasks like sentiment analysis and subjective/objective classification. TREC [20] is a question classification dataset with six distinct classes. It is widely used in information retrieval and natural language understanding research. PC [9] is a binary sentiment classification dataset that includes Pros and Cons data. It offers a unique perspective on sentiment analysis, focusing on both positive and negative aspects. The CR dataset [7] consists of customer reviews labeled as positive or negative. It provides valuable data for sentiment analysis and opinion mining. The detailed dataset statistics is shown in Table 1.

### 5.1.2 Experiment Settings.
We compare our DualCL model and its variants with existing contrastive learning baselines. We take into account the following models for comparison:
**CE:** A basic model with additional *learnable* classifiers, which is trained with the cross-entropy loss.
**CE+CL:** A contrastive learning model that jointly train the cross-entropy loss and self-supervised contrastive learning loss [11].
**CE+SCL:** The SOTA SCL model [12] which jointly train the cross-entropy loss and the supervised contrastive learning loss [17].
**DualCL w/o $\mathcal{L}_{\text{dual}}$:** The ablated model of DualCL that only trains with $\mathcal{L}_{\text{CE}}$ and incorporates proposed external data augmentations. The difference compared to CE is that the classifier of this model is *generated* from PLM without additional parameters.
**DualCL:** The proposed framework that jointly optimizes the dual contrastive learning loss $\mathcal{L}_{\text{Dual}}$ and the cross-entropy loss $\mathcal{L}_{\text{CE}}$.
**DualCL+HE, DualCL+SE:** DualCL+HE and DualCL+SE are DualCL models adopting hard ensembling (HE) and soft ensembling (SE) of the generated classifiers, respectively.

To make a fair comparison, we reproduce the results of all baselines with the same hyperparameter configurations as in the original paper and report the mean accuracy averaged over 10 runs with different random seeds. To evaluate the models in low-resource scenarios, we also train the models on 10% of the training data and report the results, besides training them on the full training data.

**Table 2: Text classification results on the SST-2, SUBJ, TREC, PC and CR datasets. The models are trained with 10% of the training data or with full training data. We reproduce the results with the same hyperparameter configurations for all baselines for a fair comparison and report the average accuracy and standard deviation across 10 different random seeds.**

| Model | Method | SST-2 | SUBJ | TREC | PC | CR | Avg. |
|-------|--------|-------|------|------|-----|-----|------|
| | | | | 10% training data | | | |
| | CE | 86.05±0.24 | 93.05±0.25 | 93.29±0.22 | 91.09±0.23 | 86.58±0.29 | 90.01±0.25 |
| | CE+SCL | 86.64±0.17 | 93.20±0.16 | 93.70±0.24 | 91.46±0.23 | 88.14±0.26 | 90.63±0.21 |
| BERT | CE+CL | 87.66±0.28 | 94.27±0.21 | 94.20±0.29 | 91.67±0.27 | 87.72±0.32 | 91.10±0.27 |
| | **DualCL w/o** $\mathcal{L}_{\text{Dual}}$ | 87.90±0.19 | 93.50±0.18 | 94.01±0.31 | 91.83±0.22 | 88.13±0.30 | 91.07±0.24 |
| | **DualCL** | **88.40±0.20** | **94.50±0.21** | **94.93±0.23** | **92.36±0.16** | **89.01±0.28** | **91.84±0.22** |
| | CE | 90.91±0.23 | 94.03±0.19 | 94.51±0.21 | 90.65±0.20 | 92.06±0.27 | 92.43±0.22 |
| | CE+SCL | 91.00±0.29 | 94.37±0.30 | 94.85±0.24 | 90.82±0.20 | 92.32±0.25 | 92.67±0.26 |
| RoBERTa | CE+CL | 91.04±0.17 | 94.47±0.19 | **95.68±0.26** | 91.90±0.14 | 92.55±0.28 | 93.13±0.21 |
| | **DualCL w/o** $\mathcal{L}_{\text{Dual}}$ | 92.48±0.18 | 94.40±0.17 | 95.18±0.16 | 91.50±0.14 | 92.88±0.20 | 93.29±0.17 |
| | **DualCL** | **92.67±0.21** | **94.78±0.19** | 95.36±0.18 | **92.17±0.20** | **93.24±0.24** | **93.64±0.20** |
| | | | | full training data | | | |
| | CE | 91.19±0.23 | 96.40±0.19 | 97.21±0.20 | 95.06±0.14 | 92.09±0.24 | 94.39±0.20 |
| | CE+SCL | 91.71±0.20 | 96.25±0.19 | 97.58±0.16 | 95.26±0.13 | 93.06±0.20 | 94.77±0.18 |
| BERT | CE+CL | 91.95±0.22 | 96.72±0.15 | 97.80±0.14 | 95.21±0.11 | 93.19±0.19 | 94.97±0.16 |
| | **DualCL w/o** $\mathcal{L}_{\text{Dual}}$ | 91.99±0.15 | 96.78±0.13 | 97.70±0.19 | 95.30±0.15 | 93.14±0.19 | 94.97±0.16 |
| | **DualCL** | **92.40±0.17** | **97.20±0.17** | **98.22±0.17** | **95.56±0.14** | **93.78±0.17** | **95.43±0.16** |
| | CE | 94.09±0.24 | 96.60±0.21 | 97.10±0.20 | 95.10±0.19 | 93.41±0.24 | 95.26±0.22 |
| | CE+SCL | 93.65±0.20 | 96.73±0.23 | 97.18±0.19 | 95.35±0.19 | 93.60±0.17 | 95.30±0.20 |
| RoBERTa | CE+CL | 94.33±0.21 | 97.04±0.17 | **97.52±0.15** | 95.32±0.10 | 93.49±0.25 | 95.54±0.18 |
| | **DualCL w/o** $\mathcal{L}_{\text{Dual}}$ | 94.41±0.23 | 96.79±0.24 | 97.10±0.25 | 95.30±0.12 | 94.01±0.25 | 95.52±0.22 |
| | **DualCL** | **94.91±0.17** | **97.34±0.19** | 97.40±0.17 | **95.59±0.12** | **94.39±0.23** | **95.93±0.18** |

## 5.2 Implementation Details

We implement our model using the PyTorch deep learning framework. The code and data are released with an anonymous link for reproduction[1]. We leverage the powerful BERT-base-uncased and RoBERTa-base models as the encoders (referred to as $f$) to obtain representations. To address the potential influence of label orders during training, we adopt a random ordering strategy on the token list before feeding it into the PLM encoder. During optimization, we employ the AdamW optimizer with default settings, which include a weight decay rate of 0.01. The training process consists of a maximum of 30 epochs, with a linear learning rate decay from $2 \times 10^{-5}$ to $10^{-5}$, ensuring a gradual adjustment of the learning rate over time. To prevent overfitting, we incorporate dropout with a rate of 0.1 across all layers of the models. Additionally, a batch size of 64 is used for all datasets. We employ a grid search strategy to identify the best configuration of the hyperparameters. Through this approach, we determine that the optimal loss weight is set to $\lambda = 0.5$, striking a balance between different components of the training objective. Moreover, we set the temperature factor in contrastive learning to $\tau = 0.1$ to govern the scale of the inner products. To ensure fair execution and training, all involved experiments are conducted on an NVIDIA Tesla A100 GPU with 40GB of memory.

[1]Anonymous link for code and data: https://file.io/rKk80kR2TR3r

## 5.3 Evaluation Results on Benchmark Datasets

The evaluation results on benchmark datasets are presented in Table 2, it becomes evident that the DualCL approach, utilizing both BERT and RoBERTa encoders, consistently achieves superior classification performance across almost all experimental settings, except when the DualCL framework is employed in pre-trained RoBERTa on the TREC dataset. When compared to the CE+CL method with full training data, DualCL exhibits an average improvement of 0.46% and 0.39% on BERT and RoBERTa, respectively. These results manifest the effectiveness of our DualCL.

What's more intriguing is that DualCL demonstrates even more pronounced advantages over the CE+CL method when only 10% of the training data is available. In such cases, DualCL surpasses the CE+CL method by a notably larger margin, achieving improvements of 0.74% and 0.51% on BERT and RoBERTa, respectively. These compelling findings serve as robust evidence that the DualCL approach is highly effective, especially in the face of the inherent challenge posed by the limited availability of training data.

Moreover, the superior performance exhibited by DualCL serves as a testament to the efficacy of the parameter-free, augmentation-easy and label-aware principles and the dual contrastive learning strategy. These findings highlight the significance of incorporating these innovative ideas to enhance text classification outcomes.

**Table 3: Accuracy on the GLUE validation set. The models are trained with 10% of the training data or with full training data. We report average accuracy across 10 different random seeds. The training speed is shown through updates per second (ups/sec).**

| Model | Method | SST-2 | CoLA | MRPC | QNLI | MNLI | Avg. | ups/sec |
|---|---|---|---|---|---|---|---|---|
| | | | | 10% training data | | | | |
| BERT | CE | 90.25±0.31 | 78.62±0.22 | 71.32±0.28 | 83.10±0.29 | 77.62±0.17 | 80.18±0.25 | **12.44** |
| | DualCL | 90.78±0.29 | 78.66±0.31 | 73.49±0.36 | 85.99±0.24 | 78.08±0.20 | 81.40±0.28 | 11.32 |
| | **DualCL+HE** | 91.06±0.24 | 78.78±0.37 | 74.02±0.30 | **86.15±0.28** | 78.37±0.24 | 81.68±0.29 | 10.76 |
| | **DualCL+SE** | **91.17±0.35** | **78.83±0.34** | **74.10±0.39** | 86.09±0.31 | **78.44±0.23** | **81.73±0.32** | 10.55 |
| RoBERTa | CE | 92.78±0.29 | 78.81±0.39 | 78.92±0.23 | **88.31±0.25** | **83.20±0.21** | 84.40±0.27 | **11.25** |
| | DualCL | 92.91±0.22 | 78.95±0.36 | 79.59±0.27 | 87.63±0.30 | 83.04±0.19 | 84.42±0.27 | 10.94 |
| | **DualCL+HE** | 92.94±0.34 | **79.03±0.27** | 80.13±0.31 | 87.75±0.33 | 83.04±0.17 | 84.58±0.28 | 10.81 |
| | **DualCL+SE** | **92.99±0.26** | 78.93±0.24 | **80.22±0.28** | 88.01±0.34 | 83.14±0.26 | **84.66±0.28** | 10.14 |
| | | | | full training data | | | | |
| BERT | CE | 92.32±0.17 | 83.03±0.22 | 81.37±0.24 | 91.07±0.23 | 83.83±0.37 | 86.32±0.25 | **12.42** |
| | DualCL | 92.44±0.29 | 83.20±0.22 | 83.89±0.27 | 91.24±0.21 | 84.03±0.31 | 86.96±0.26 | 11.26 |
| | **DualCL+HE** | 92.53±0.31 | 83.32±0.28 | **84.80±0.37** | 91.35±0.26 | 84.17±0.21 | 87.23±0.29 | 10.88 |
| | **DualCL+SE** | **92.78±0.33** | **83.46±0.24** | 84.75±0.30 | **91.39±0.28** | **84.24±0.32** | **87.32±0.29** | 10.49 |
| RoBERTa | CE | 94.15±0.19 | **84.08±0.24** | 87.99±0.18 | 92.73±0.21 | 87.04±0.26 | 89.20±0.22 | **11.20** |
| | DualCL | 94.20±0.25 | 83.83±0.27 | 88.27±0.29 | 92.76±0.22 | 87.04±0.19 | 89.22±0.24 | 11.21 |
| | **DualCL+HE** | 94.38±0.30 | 83.99±0.32 | 88.48±0.35 | **92.84±0.33** | **87.07±0.20** | 89.35±0.30 | 10.89 |
| | **DualCL+SE** | **94.54±0.22** | 83.97±0.29 | **88.56±0.30** | 92.84±0.35 | 87.09±0.24 | **89.40±0.28** | 10.07 |

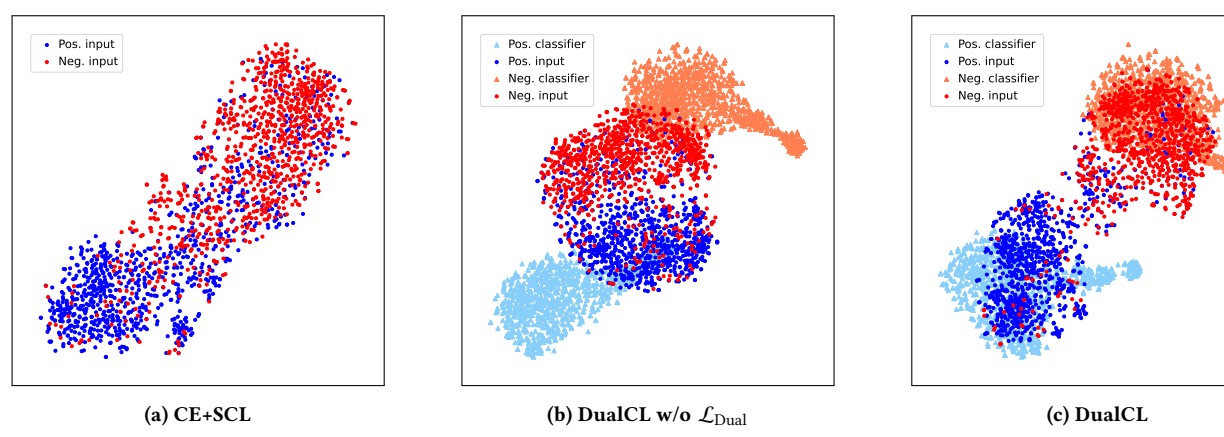

(a) CE+SCL                     (b) DualCL w/o $\mathcal{L}_{\text{Dual}}$                     (c) DualCL

**Figure 3: The tSNE plots of the learned representations on the SST-2 dataset.**

## 5.4 Visualization of Learned Representations

To delve into the investigation of how dual contrastive learning enhances the quality of representations, we present the t-SNE plots that showcase the learned representations on the SST-2 test set. For our experimentation, we employ RoBERTa as the encoder and proceed to fine-tune the encoder using 25 training samples for each class. In Figure 3, we visualize the outcomes of three distinct approaches: CE, DualCL w/o $\mathcal{L}_{\text{Dual}}$, and DualCL.

Upon analyzing the t-SNE plots in Figure 3, we expect to investigate the model's capability to produce discriminative representations for the input samples and the classifier parameters associated with each example. By comparing Figure 3 (c) with Figure 3 (a) and Figure 3 (b), we observe a noticeable distinction. Specifically,

the representations of the input texts and the corresponding class parameters are better aligned with each other in DualCL than in CE+SCL and DualCL w/o $\mathcal{L}_{\text{Dual}}$. It becomes evident that the dual contrastive loss plays a crucial role in facilitating the model's acquisition of more discriminative and robust representations for both the input representations and the generated classifier parameters.

The t-SNE plots depicted in Figure 3 suggest the effectiveness of the improved learning process of DualCL. In essence, by integrating the dual contrastive loss into the DualCL framework, the model becomes capable of leveraging the inherent duality between training examples and classifier parameters. This integration enables the model to generate more insightful and robust representations for both the input samples and the classifier parameters.

## 5.5 Effectiveness of Ensembled Classification

To validate the effectiveness of the ensembled classification approach on more challenging datasets, we conduct experiments on the subset of GLUE benchmark tasks [31] (SST-2, CoLA, MRPC, QNLI and MNLI) in both full-data and low-resource settings. The results obtained are presented in Table 3. Notably, we observed that DualCL+HE and DualCL+SE, when incorporating ensembled predictions, almost consistently outperform their non-ensembled counterparts. Specifically, in comparison to the DualCL method, DualCL+HE with hard ensemble exhibited an improvement of 0.28% and 0.15% in accuracy for the BERT and RoBERTa models, respectively. Furthermore, DualCL+SE with soft ensemble showcased an additional enhancement of 0.35% for the BERT model and 0.21% for the RoBERTa model when compared to the DualCL method.

The above results suggest that the ensembled classification approach is effective in improving text classification performance by aggregating multiple outputs. It is worth mentioning that the computational overheads introduced by the ensemble-based prediction schemes were minimal, as evident from the last column of Table 3. Although CE is known for its exceptional training speed, the DualCL approach and its ensembled variants demonstrate a comparable training speed, suffering only negligible speed loss.

## 5.6 Effectiveness in Low-Resource Scenarios

In order to examine the effectiveness of the proposed dual contrastive loss and the data augmentations generated by internal components in enhancing text classification performance under low-resource scenarios, we conducted experiments on the SST-2 and SUBJ datasets with varying numbers of training examples. Specifically, we evaluated the performance of the compared models using different numbers of $N$ training samples per class, where $N$ is selected from the set of {5, 10, 20, 30, 40, 50, 60, 70, 80, 90, 100}.

To present the findings, we depicted the results of three models: BERT trained with cross-entropy loss (CE), DualCL without the dual contrastive learning loss (DualCL w/o $\mathcal{L}_{\text{Dual}}$), and DualCL. The test accuracy of using a various number of training examples on the SST-2 and SUBJ datasets is shown in Figure 4. The evaluation results clearly demonstrate that DualCL outperforms CE and DualCL without $\mathcal{L}_{\text{Dual}}$, especially when the number of training examples is limited. The improvement achieved by DualCL is particularly noteworthy, reaching up to about 8.5% on SST-2 and 5.4% on SUBJ, even when only 5 training samples are available per class.

It is worth noting that even without utilizing the dual contrastive loss, DualCL w/o $\mathcal{L}_{\text{Dual}}$, which is trained using cross-entropy loss with the generated classifiers and data augmentations, consistently outperforms CE in low-resource scenarios. This observation underscores the efficacy of the proposed internal data augmentations as a means to improve the generalization capability of the model, which leads to better classification results when training data is scarce.

## 5.7 Case Study

To study the effectiveness of DualCL in capturing informative features, we employ an attention-scoring mechanism between the representation of the "[CLS]" token and each word in the given sentence. Initially, we fine-tuned the RoBERTa encoder using the complete training set. Subsequently, we compute the $\ell_2$ distance

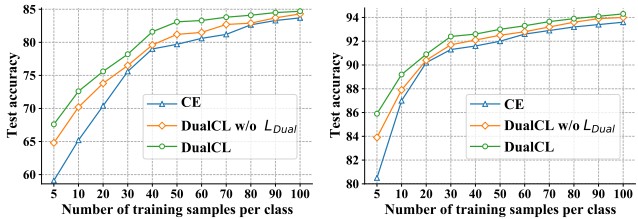

**Figure 4: Test accuracy on the SST-2 (left) and SUBJ (right) datasets with different numbers of training examples.**

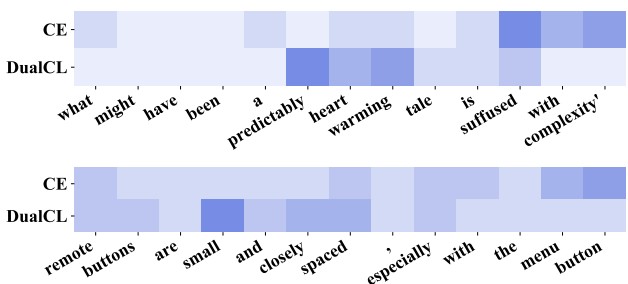

**Figure 5: The visualization of attention map for CE and DualCL. The darker blue refers to higher attention scores.**

between the extracted features and proceed to visualize the resulting attention map in Figure 5. From the visualization results, we can conclude that the attention maps of different words exhibit distinguishable patterns when performing the classification.

For instance, in the upper example sampled from the SST-2 dataset, our model demonstrates higher attention towards the phrase "predictably heart warming" in the sentence representing a "positive" sentiment. Conversely, in the lower example derived from the CR dataset, our DualCL model emphasizes the word "small" with greater attention, indicative of "negative" sentiment. In contrast, the CE method fails to concentrate on these distinctive features. These outcomes indicate that our DualCL framework successfully learns to attend to the informative keywords within the sentence, enabling it to capture relevant and discriminative features effectively.

## 6 CONCLUSION

In this paper, we attempt to overcome the potential shortcomings in the existing supervised contrastive learning (SCL) approach for text classification. We propose the parameter-free, augmentation-easy and label-aware principles to design an effective SCL approach. Motivated by our derived lower bound of mutual information maximization based on the duality of text classification, we develop a dual contrastive learning framework DualCL. This framework generates classifier parameters as augmented views of the input text and simultaneously uses them for ensembled classification. Extensive experimentation convincingly demonstrates the efficacy of both the proposed principles and the DualCL framework. We aspire for this streamlined and efficient framework to emerge as a compelling alternative to existing supervised contrastive learning methods, demonstrating its versatility across various domains.

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

# APPENDIX

## 6.1 Proof of Theorem 1

THEOREM 1. *Assume that there is a constant $\epsilon$ such that $p(\boldsymbol{x}_i, y_i) \geq \epsilon$ holds for all $i \in \mathcal{I}$ and $\frac{p(y_j|\boldsymbol{x}_i)}{p(y_j)} \propto \phi(\boldsymbol{x}_i, y_j)$:*

$$\mathrm{MI}(\mathcal{X}, \mathcal{Y}) \geq \log N - \epsilon(\mathcal{L}_{\boldsymbol{\theta} \leftarrow z} + \mathcal{L}_{z \leftarrow \boldsymbol{\theta}}) \tag{11}$$

*where $\phi$ is a symmetric function with diverse definitions. In our case, $\phi(\boldsymbol{x}_i, y_j) = (g(\boldsymbol{\theta}_i^*, z_j) + g(\boldsymbol{\theta}_j^*, z_i))/2$.*

PROOF. Let $M_i = \sum_{j=1}^{N} \frac{p(y_j|\boldsymbol{x}_i)}{p(y_j)}$ and assume that $\frac{1}{|\mathcal{P}_i|} \sum_{p \in \mathcal{P}_i} \phi(\boldsymbol{x}_i, y_p) = \phi(\boldsymbol{x}_i, y_i)$ when $|\mathcal{P}_i|$ is sufficiently large. We have:

$$\mathrm{MI}(\mathcal{X}, \mathcal{Y}) = \frac{1}{N} \sum_{i=1}^{N} \sum_{j=1}^{N} p(\boldsymbol{x}_i, y_j) \log \left( \frac{p(y_j|\boldsymbol{x}_i)}{p(y_j)} \right)$$

$$= \frac{1}{N} \sum_{i=1}^{N} \sum_{j=1}^{N} p(\boldsymbol{x}_i, y_j) \left( \log \frac{p(y_j|\boldsymbol{x}_i)}{p(y_j)M_i} + \log M_i \right)$$

$$= \frac{1}{N} p(\boldsymbol{x}_i, y_i) \log \frac{\phi(\boldsymbol{x}_i, y_i)}{\sum_{t=1}^{N} \phi(\boldsymbol{x}_i, y_t)}$$

$$+ \frac{1}{N} \sum_{j \neq i} p(\boldsymbol{x}_i, y_j) \log \frac{\phi(\boldsymbol{x}_i, y_j)}{\sum_{t=1}^{N} \phi(\boldsymbol{x}_i, y_t)} + \log N$$

$$\geq \log N + \frac{\epsilon}{N} \sum_{i=1}^{N} \log \frac{\phi(\boldsymbol{x}_i, y_i)}{\sum_{t=1}^{N} \phi(\boldsymbol{x}_i, y_t)} \tag{12}$$

$$= \log N + \frac{\epsilon}{N} \sum_{i=1}^{N} \frac{1}{|\mathcal{P}_i|} \sum_{p \in \mathcal{P}_i} \log \frac{\phi(\boldsymbol{x}_i, y_p)}{\sum_{t=1}^{N} \phi(\boldsymbol{x}_i, y_t)}$$

$$\geq \log N + \frac{\epsilon}{N} \sum_{i=1}^{N} \log \frac{\phi(\boldsymbol{x}_i, y_i)}{\sum_{t=1}^{N} \phi(\boldsymbol{x}_i, y_t)}$$

$$\geq \log N + \frac{\epsilon}{N} \sum_{i=1}^{N} \frac{1}{|\mathcal{P}_i|} \sum_{p \in \mathcal{P}_i} \log \frac{g(\boldsymbol{\theta}_i^*, z_p)}{\sum_{a=1}^{N} g(\boldsymbol{\theta}_i^*, z_a)}$$

$$+ \frac{\epsilon}{N} \sum_{i=1}^{N} \frac{1}{|\mathcal{P}_i|} \sum_{p \in \mathcal{P}_i} \log \frac{g(\boldsymbol{\theta}_p^*, z_i)}{\sum_{a=1}^{N} g(\boldsymbol{\theta}_a^*, z_i)}$$

$$= \log N - \epsilon(\mathcal{L}_{\boldsymbol{\theta} \leftarrow z} + \mathcal{L}_{z \leftarrow \boldsymbol{\theta}})$$

□

This theorem proves that the negative of loss $\mathcal{L}_{\boldsymbol{\theta} \leftarrow z} + \mathcal{L}_{z \leftarrow \boldsymbol{\theta}}$ is a lower bound of the mutual information $\mathrm{MI}(\mathcal{X}, \mathcal{Y})$. Thus, when we specify the loss $\mathcal{L}_{\boldsymbol{\theta} \leftarrow z} + \mathcal{L}_{z \leftarrow \boldsymbol{\theta}}$ by the dual contrastive loss $\mathcal{L}_{\mathrm{Dual}}$ and minimize it, the mutual information between inputs and labels is accordingly maximized.

