# OpenReview forum: "DualCL: Principled Supervised Contrastive Learning as Mutual Information Maximization for Text Classification"
_ACM.org/TheWebConf/2024/Conference — TheWebConf24_

### Official Review · Reviewer_yr7D · 2023-11-23

**Novelty:** 3
**Technical Quality:** 4

**Review:**

This paper targets the classical problem of text classification. The authors propose a principled supervised contrastive learning model, which is parameter-free, augmentation-easy, and label-aware. This model, DualCL, can effectively capture the mutual information text representations and classifier parameters. Meanwhile, the authors also provide relevant theoretical support. The outstanding performance on various datasets demonstrates the effectiveness of the proposed model.

**Strengths**
1. The model description section is well-written.

2. The experimental results presented in the paper are convincing.

3. They have demonstrated the method's efficacy in terms of performance. Comparison with state-of-the-art methods further underscores the robustness and superiority of the proposed approach.

**Weaknesses**
1. Can the authors provide reasons why we still need this kind of supervised contrastive learning model in the era of large language models?

2. From the results in the table, the improvement brought by the model does not seem significant. Additionally, most of the datasets are binary classification. Therefore, can the model still perform well on datasets with many categories?

**Questions:**

See the weaknesses.

**Reviewer Confidence:**

4: The reviewer is certain that the evaluation is correct and very familiar with the relevant literature

**Scope:**

3: The work is somewhat relevant to the Web and to the track, and is of narrow interest to a sub-community

---

### Official Review · Reviewer_PBFv · 2023-11-30

**Novelty:** 4
**Technical Quality:** 5

**Review:**

This paper presents a new method for using PLM + supervised contrastive learning for text classification. It first critiques the current supervised contrastive learning (SCL) methods for lacking in training effectiveness and ease of deployment. To address these issues, the authors propose three design principles: parameter-free, augmentation-easy, and label-aware. They introduce DualCL, a novel dual contrastive learning framework, which leverages a derived mutual information maximization lower bound. The authors claim that DualCL can generate classifier parameters via PLMs, using these for both classification and as augmented text views for SCL. Extensive experiments across various text classification datasets show DualCL's superior performance in learning text representations and its significant improvement over baseline models.

Overall this paper is clearly written, easy to digest, and the presented framework is reasonable. The derivation of MI lower bound and the “trick” of “augmentation-easy implementation” (namely prepending the class tokens in front of the text sequence) are particularly interesting. Therefore I think overall the paper is worth reading.

However, I still think the paper has multiple places to improve.

First, the motivation of the new framework is not very clear. The authors claim the old methods have three limitations (around line 310) but I don’t quite buy the first two shortcomings. From eq. 1, I don’t see a classifier parameter theta as additional parameters to train and there is no mention of “external text data augmentation”. Further explanation for the first two shortcomings are needed.

Second, I feel the concept of “duality” is not clearly described, around line 341, the authors mention ” “When training this model, the softmax transformation of inner product \theta* times z_i is forced to be maximized through the optimizer. This forms a dual relationship between the text representations and the classifier parameters.” Why does the dot product mean the dual relationship? Because of the symmetricity of \theta* and z_i?

Third, I think the experiments can be further improved. Main experiment datasets are mostly about sentiment classification, what about more large-scale topic classification? Plus, the baselines seem weak, mostly PLM combined with different loss functions and there is no specifically designed text classification model. It would be better to try more recent classification models.

Finally, I feel the connection of MI and text classification accuracy can be further explained. Why maximize the lower bound of MI (or even the MI itself) can definitely lead to better classification accuracy? What if those classes have many semantic correlations? Some discussions here can be helpful.

**Questions:**

1. Have you tried your method on more recent seq2seq based classification model (e.g., t5, PaLM, or finetune Llama)?
2. Are the performance improvements in table 1 and 3 statistically significant?
3. Is the gap between your exact MI and MI lower bound huge? Can you compute the gap on a small dataset and show that lower bound is meaningful?

**Reviewer Confidence:**

2: The reviewer is willing to defend the evaluation, but it is likely that the reviewer did not understand parts of the paper

**Scope:**

4: The work is relevant to the Web and to the track, and is of broad interest to the community

---

### Official Review · Reviewer_HiXw · 2023-12-01

**Novelty:** 4
**Technical Quality:** 5

**Review:**

**Summary**

This paper proposed three main principles for effective training and deployment of friendliness in a supervised contrastive learning approach for text classification. Based on these principles, this paper proposed DualCL, a supervised contrastive learning framework.
By augmenting the classifier parameters, the framework introduces no learnable parameters, easily augments anchor samples, and leverages class semantics to enhance the representations. Extensive experimental results on various datasets are provided to describe the performance of the proposed method in comparison to other SCL models.


**Strength**
* The idea of augmenting classifier parameters is interesting.
* The paper introduces a singular model design that handles three limitations in existing SCL models simultaneously by augmenting classifier parameters: achieving parameter-free, augmentation-easy, and label-awareness.
* The proposed method is well-grounded in theoretical analysis.
* The paper evaluates the proposed method extensively including different two tasks and various scenarios with various types of datasets.

**Weaknesses**
*  Definition of principles: While the paper “proposed” three key principles for designing the SCL model in terms of training effectiveness and deployment-friendliness, a more solid discussion about them is needed.
    The reasons behind their necessity require further explanation.
    * For example, in the context of SCL models for text classification tasks, the significance of the parameter-free principle needs a more compelling explanation or examples.
* Incremental contribution of the proposed model compared to LaCon: Based on three principles, the difference between LaCon and the proposed model lines in LaCon’s introduction of learnable parameters.
    * The 224th line in Section 2 explains the difference between LaCon and the proposed model: “ ….LaCon introduces a learnable matrix with extra parameters to look up label embeddings…..”
    * Could authors provide a more detailed explanation of how critical this difference is?

* The proposed method showed marginal increases in accuracy (less than 1%) even though the method consistently outperforms baselines.

**Questions:**

* Which existing methods satisfy these principles or not?
    * Providing a more **organized** comparative analysis of methods would be helpful. For example, It would be easier to compare existing methods and the proposed model by putting a table that compares existing methods and the proposed method in terms of three principles even in the appendix if there is a concern about the limitation of space.

* Are there any downsides to training a classifier other than the downside that introduces new trainable parameters?
    * Could authors explain more about why introducing additional parameters can be a main shortcoming. Using fewer parameters will be good. However, if the classifier is not big enough, it is not a critical problem in terms of training.
* Are there any reasons to design generating classifier parameters as explained in the paper?
    * Could authors explain more about the reason behind the design choice to generate classifier parameters in terms of benefits or drawbacks in capturing accurate class semantics except for the benefit that it can directly generate embeddings?
* How do other SCL methods behave in the limited resource scenario?
* Why is Lacon not compared to the proposed method?

**Mistakes**
* The provided link for the source code and datasets is invalid.
* Check the references in Section 1: …sentimental analysis [27] and review classification [23] …
* Correct the LaTeX formatting in Section 2. Use \subsection{} instead of \subsubsection{}.
* There is a mismatch between a Figure 3 (a) label (CE+SCL) and an explanation (CE, 748th) in Section 5.4

**Ethics Review Description:**

There are no ethical issues.

**Reviewer Confidence:**

2: The reviewer is willing to defend the evaluation, but it is likely that the reviewer did not understand parts of the paper

**Scope:**

3: The work is somewhat relevant to the Web and to the track, and is of narrow interest to a sub-community

---

### Decision · Program_Chairs · 2024-01-22

**Decision:**

Accept

**Comment:**

The paper introduces Dual Contrastive Learning (DualCL) for text classification, a novel framework designed to enhance performance by optimizing both the feature space and the classifier space. This method addresses a key limitation in supervised contrastive learning by integrating classifier parameters into the contrastive loss function.

 Strengths:
 1. **Theoretical Rigor**: DualCL is grounded in a principled approach, blending information theory with practical machine learning.
 2. **Enhanced Accuracy**: Demonstrates improved classification accuracy across multiple datasets, indicating its effectiveness in diverse contexts.
 3. **Dual Optimization**: Unique in optimizing both feature and classifier spaces, leading to more robust and discriminative representations.

 Weaknesses:
 1. **Scalability and Generalizability**: While effective on tested datasets, its scalability to larger, more diverse datasets and its generalizability to other domains are not thoroughly explored.


 The reviewers raised several concerns about the paper, with the main concerns being:

 1. **Clarity and Justification of Principles**: Reviewers sought more solid discussions on the three proposed principles for designing the SCL model, questioning their necessity and significance.
 2. **Comparison with LaCon**: There was a request for a clearer explanation of the critical differences between DualCL and LaCon, especially regarding their contributions.
 3. **Marginal Performance Increases**: Concerns were raised about the relatively small performance improvements of DualCL over baseline models.
 4. **Experimental Scope**: Suggestions were made to broaden the range of experiments and include more recent text classification models.
 5. **Connection Between MI and Classification Accuracy**: Reviewers requested further explanation on how maximizing mutual information (MI) leads to better classification accuracy.

 The authors provided detailed explanations to the reviewer concerns. In summary, DualCL represents a significant advancement in text classification through its unique dual optimization approach.